# Flap Endonuclease 1 Endonucleolytically Processes RNA to Resolve R-Loops through DNA Base Excision Repair

**DOI:** 10.3390/genes14010098

**Published:** 2022-12-29

**Authors:** Eduardo E. Laverde, Aris A. Polyzos, Pawlos P. Tsegay, Mohammad Shaver, Joshua D. Hutcheson, Lata Balakrishnan, Cynthia T. McMurray, Yuan Liu

**Affiliations:** 1Biochemistry Ph.D. Program, Florida International University, Miami, FL 33199, USA; 2Molecular Biophysics and Integrated Bioimaging, Lawrence Berkeley National Laboratory, Berkeley, CA 94720, USA; 3Department of Biomedical Engineering, Florida International University, Miami, FL 33199, USA; 4Department of Biology, Indiana-Purdue University, Indianapolis, IN 46202, USA; 5Department of Chemistry and Biochemistry, Florida International University, Miami, FL 33199, USA; 6Biomolecular Sciences Institute, Florida International University, Miami, FL 33199, USA

**Keywords:** flap endonuclease 1 (FEN1), R-loop, base excision repair (BER)

## Abstract

Flap endonuclease 1 (FEN1) is an essential enzyme that removes RNA primers and base lesions during DNA lagging strand maturation and long-patch base excision repair (BER). It plays a crucial role in maintaining genome stability and integrity. FEN1 is also implicated in RNA processing and biogenesis. A recent study from our group has shown that FEN1 is involved in trinucleotide repeat deletion by processing the RNA strand in R-loops through BER, further suggesting that the enzyme can modulate genome stability by facilitating the resolution of R-loops. However, it remains unknown how FEN1 can process RNA to resolve an R-loop. In this study, we examined the FEN1 cleavage activity on the RNA:DNA hybrid intermediates generated during DNA lagging strand processing and BER in R-loops. We found that both human and yeast FEN1 efficiently cleaved an RNA flap in the intermediates using its endonuclease activity. We further demonstrated that FEN1 was recruited to R-loops in normal human fibroblasts and senataxin-deficient (AOA2) fibroblasts, and its R-loop recruitment was significantly increased by oxidative DNA damage. We showed that FEN1 specifically employed its endonucleolytic cleavage activity to remove the RNA strand in an R-loop during BER. We found that FEN1 coordinated its DNA and RNA endonucleolytic cleavage activity with the 3′-5′ exonuclease of APE1 to resolve the R-loop. Our results further suggest that FEN1 employed its unique tracking mechanism to endonucleolytically cleave the RNA strand in an R-loop by coordinating with other BER enzymes and cofactors during BER. Our study provides the first evidence that FEN1 endonucleolytic cleavage can result in the resolution of R-loops via the BER pathway, thereby maintaining genome integrity.

## 1. Introduction

Flap endonuclease 1 (FEN1) is a central component of DNA metabolism that plays an essential role in removing RNA primers and DNA base damage during DNA lagging strand maturation and long-patch base excision repair (BER) [1,2]. FEN1 belongs to the RAD2 structure-specific nuclease superfamily. It contains endo- and exonucleolytic cleavage activity and gap endonuclease activity [3,4,5,6]. During DNA replication, the DNA lagging strand is initially processed by RNase H1, which removes RNA primers, leaving a multi-nucleotide gap with the last ribonucleotide attached to a downstream DNA strand [1,2]. Subsequently, DNA polymerase δ (pol δ) fills in the gap and strand-displaces the downstream DNA generating a 5′-flap with the ribonucleotide [1,2]. FEN1 then endonucleolytically cleaves the flap removing the last ribonucleotide and generating a nick that is sealed by DNA ligase I (LIG I) [1,2]. FEN1 can use a “tracking mechanism” [7] during which the enzyme is loaded from the 5′-end of the flap, tracks along the flap down to the junction between the flap and annealed region, and cleave the flap. It can also employ a “flap threading mechanism” through which it initially binds to the junction of the flap, threads the flap through itself, and cleaves the flap [8,9]. The unique mechanisms of FEN1 flap cleavage make it a versatile enzyme that can efficiently cleave a 5′-flap on a variety of DNA replication and repair intermediates. Furthermore, FEN1 also plays a crucial role in preventing sequence duplication, repeat sequence expansion, and telomere instability and fragility [1,2,10,11]. The important roles of FEN1 in maintaining genome stability and integrity have been further demonstrated by the fact that the insufficiency of FEN1 is associated with lung and gastrointestinal cancers [12,13] and induces various mutations and genome instability in cancers [14,15].

Besides its roles in DNA replication, BER, and genome maintenance, FEN1 is involved in other cellular functions by processing DNA and RNA and modulating epigenetic features. It is implicated that FEN1 can induce apoptosis by interacting with the proteins of DNA degradosome [16]. The enzyme is also involved in the formation of covalently closed circular (cccDNA) in the hepatitis B virus [17] and cell morphogenesis [18]. In addition, FEN1 can modify the epigenetic features in cancer cells by inducing the upregulation of DNA methyltransferase 1 (DNMT1) and DNMT3a and interacting with DNMT3a through proliferating cell nuclear antigen (PCNA) [19]. Consequently, this facilitates DNA hypermethylation at the promoter region of microRNA-200a (miRNA-200a), stimulating the expression of hepatocyte growth factor (HGF) and epidermal growth factor receptor (EGFR) and promoting breast cancer progression [19].

Interestingly, FEN1 is also implicated in RNA processing and ribosome biogenesis [18]. An early study from the Bambara group has shown that calf thymus FEN1 can cleave an RNA flap endonucleolytically during DNA lagging strand processing [20]. A later study shows that human FEN1 can make a cleavage on mRNA and rRNA endonucleolytically [21]. However, the biological significance of FEN1 RNA processing remains unknown. A recent study has suggested that FEN1 may be involved in RNA processing in R-loops, formed at the telomeres of the leading strand DNA, alleviating telomere fragility [11]. However, it remains unknown how FEN1 can process an RNA strand in an R-loop. We have recently discovered that FEN1 can cleave the RNA strand on a trinucleotide repeat (TNR) R-loop leading to repeat deletion [22] through BER, suggesting a unique mechanism for FEN1 to resolve R-loops.

R-loops are bulky non-B form DNA structures that contain an RNA:DNA hybrid and a non-template single-strand DNA (ssDNA), frequently generated during gene transcription [23]. They play an essential role in mediating immunoglobulin class switch recombination [24], CRISPR-Cas9-mediated DNA cleavage [25,26], mitochondrial DNA replication [27], and prevention of gene silencing by suppressing DNA methylation at CpG islands [28]. However, non-scheduled spontaneous formation of R-loops in the genome can result in a series of pathological consequences. These include replication fork stalling, blockage of mRNA synthesis, DNA damage, transcription-associated DNA recombination, and DNA repeat sequence instability, ultimately leading to cancer and neurodegeneration [23,29]. To combat these adverse effects, cells have evolved multiple mechanisms that prevent unscheduled R-loop accumulation. These include the removal of the RNA strand by RNase H, dissociation of the RNA by RNA-dependent helicases/ATPases such as senataxin (SETX) and Fanconi anemia complementation group M (FANCM), suppression of reannealing of the RNA strand to its template by DNA topoisomerases, and process of an R-loop through DNA repair [23]. However, it remains unknown how DNA repair is involved in removing R-loops and how the multiple mechanisms and their coordination can lead to the efficient resolution of R-loops. Recent findings suggest a possibility that FEN1 may cleave a non-template strand of an R-loop to induce double-strand DNA breaks in coordination with topoisomerase I (Top I) and other nucleases [30]. Depletion of FEN1 in human cells can result in the deficiency of R-loop processing, thereby promoting the accumulation of R-loops induced by topoisomerase I cleavage complex (TOP1ccs) and reducing DNA strand breaks resulting from R-loop processing [30]. It is also suggested that the single-stranded non-template strand on R-loops are susceptible to nucleases and reactive oxygen species (ROS) [23]. This may result in ssDNA breaks that are repaired by BER through which R-loops may be resolved. We further hypothesize that DNA base damage that occurs on the non-template ssDNA of R-loops induces BER during which FEN1 removes the RNA strand and a DNA flap in the R-loops through its coordination with key BER enzymes such as DNA polymerase β (pol β), leading to the removal of R-loops. To test this hypothesis, we initially examined FEN1 cleavage activity on RNAs. We then determined if FEN1 can colocalize with R-loops upon oxidative DNA base damage in human fibroblasts with or without SETX deficiency. Finally, we characterized FEN1 functional coordination with AP endonuclease 1 (APE1) and pol β in removing an RNA strand in resolving an R-loop through BER. We found that both yeast and human FEN1 cleaved RNA flaps endonucleolytically. We further demonstrated that FEN1 tracked down to the annealed DNA region to cleave a flap with RNA. We showed that FEN1 was recruited to R-loops in human cells upon oxidative DNA base damage. We demonstrated that FEN1 endonuclease and the 3′-5′ exonuclease APE1 coordinated to remove an RNA strand leading to the resolution of a R-loops through BER.

## 2. Materials and Methods

### 2.1. Materials

The primary normal human fibroblasts and ataxia with oculomotor apraxia type 2 (AOA2) patient fibroblasts with SETX deficiency were generously provided by Dr. Kenneth Fischbeck at the National Institute of Neurological Disorders and Stroke/National Institutes of Health. Fetal Bovine Serum (FBS) and Dulbecco’s Modified Medium (DMEM) high glucose cell culture medium were from Thermo Fisher Scientific (Waltham, MA, USA). 

RNA and DNA oligonucleotides were synthesized by Integrated DNA Technologies (IDT, Coralville, IA, USA) or Eurofins Genomics (Louisville, KY, USA). Radionucleotides, α-^32^P-cordycepin triphosphate (5000 Ci/mmol), and γ-^32^P-adenosine triphosphate (3000 Ci/mmol) were from PerkinElmer Inc. (Boston, MA, USA). T4 polynucleotide kinase was from New England BioLabs (Ipswich, MA, USA). Terminal deoxynucleotidyl transferase was from Thermo Fisher Scientific (Waltham, MA, USA). Diethyl pyrocarbonate (DEPC) was from MP Biomedicals (Santa Ana, CA, USA). Deoxynucleosides triphosphate were from Sigma-Aldrich (St. Louis, MO, USA). All other chemicals were from Thermo Fisher Scientific (Waltham, MA, USA) and Sigma-Aldrich (St. Louis, MO, USA).

### 2.2. Purification of Recombinant BER Enzymes

Recombinant human APE1, pol β, FEN1, and LIG I was expressed in *Escherichia coli* (*E. Coli*) BL21(DE3) and purified using FPLC according to the procedures described [31,32,33]. For all recombinant BER enzymes, two liters of bacterial cell culture were grown. The protein expression was induced at OD of 0.6 at 37 °C or 16 °C (LIG I) for 3.5 h or 16 h (LIG I) using 1 mM Isopropyl β-D-1-thiogalactopyranoside (IPTG). Bacterial cells were harvested and lysed by a French Press (GlenMills, Clifton, NJ, USA). Soluble proteins were harvested through centrifugation at 12,000× *g* rpm for 30 min. Proteins were separated through affinity, ion exchange, and hydrophobic chromatography. After a series of sequential chromatographic purification, the peak fractions of BER proteins were pooled and dialyzed into the storage buffer containing 30 mM HEPES, pH 7.5, 50 mM KCl, 20% glycerol, and 1 mM PMSF. Recombinant *Saccharomyces cerevisiae* FEN1 (Rad27) was cloned into T7 expression vector pET-24b (Novagen/EMD Biosciences, WI), expressed in *E. coli* strain BL21(DE3) codon plus (Stratagene), and purified as previously described, resulting in a recombinant protein with a C-terminal His-tag [34]. All purified recombinant proteins were stored at −80 °C.

### 2.3. Oligonucleotide Substrates

Duplex oligonucleotide substrates containing a 19 nt RNA primer were constructed by annealing a downstream primer with an RNA primer, the template, and a DNA upstream primer with varying lengths (29–48 nt) at a molar ratio of 1:3:5. The R-loop substrate containing an abasic site (tetrahydrofuran, THF, an analog of an abasic site) on the non-template DNA and 36 nt RNA:DNA hybrid was created by annealing the non-template DNA with a THF residue and the 36 nt-RNA strand with the template strand at a molar ratio of 4:1:3. The substrate that contains the double DNA flaps with a 36 nt RNA:DNA hybrid was constructed by annealing the 34 nt upstream primer, the RNA primer, the 33 nt downstream DNA strand with a THF residue with the template strand at a ratio of 4:1:4:3. Substrates were constructed in the annealing buffer (50 mM NaCl, 1 mM EDTA, 10 mM Tris, pH 7.5) that was prepared with water treated by 0.1% diethylpyrocarbonate (DEPC) with denaturation of the oligonucleotides at 96 °C for 5 min and cooling down to 25 °C. The sequences of the substrates are listed in Appendix A. Substrates were radiolabeled at the 5′-end of the RNA strand or 5′-end of the upstream strand, or the 5′ or 3′-end of the downstream strand.

### 2.4. Cleavage of RNA by BER Enzymes and BER Enzymatic Reactions

Cleavage of RNA by FEN1 and APE1 and BER enzymatic activities were determined by incubating 25 nM substrates with different concentrations of FEN1 and APE1 in the absence or presence of pol β and LIG I in BER reaction buffer (30 mM HEPES, pH 7.5, 50 mM KCl, 0.1 mg/mL BSA, 0.1 mM EDTA, and 0.01% NP-40). Reconstituted BER reactions were performed by incubating 25 nM substrate with different concentrations of FEN1 in the presence of 50 nM APE1, 5 nM pol β, and 10 nM LIG I or with various concentrations of APE1 in the presence of 25 nM FEN1, 5 nM pol β, and 10 nM LIG I. Reactions (20 µL) were assembled in BER reaction buffer containing 50 µM dNTPs, 5 mM Mg^2+^, and 2 mM ATP at 37 °C for 30 min. Reactions were stopped with a 2× stopping buffer (95% deionized formamide and 10 mM EDTA). 

### 2.5. Detection of FEN1 Recruitment to R-Loops by Immunofluorescence

#### 2.5.1. DNA Damage Induced in Human Fibroblasts

Normal human fibroblasts (Hum Fbs) and senataxin deficient human fibroblasts (AOA2 Fbs) were grown to 60−80% confluence in Media (DMEM (Life Technologies/Thermo Fisher Scientific, cat#10569044, Carlsbad, CA, USA), 20% FBS (R&D Systems, Inc/biotechne, cat#S11150, Minneapolis, MN, USA), 1 mM sodium pyruvate (VWR International/Avantor, cat# 16777-188, Radnor, PA, USA), 1 mM GlutaMax (Thermo Fisher Scientific, cat# 35050061, Waltham, MA, USA), 1× MEM-NEAA (VWR International/avantor, cat# 10128-762, Radnor, PA, USA), 1× Antibiotic-Antimycotic (Life Technologies/Thermo Fisher Scientific, cat# 15240062, Carlsbad, CA, USA) in reduced oxygen conditions (37 °C, 5% CO_2_ and 3% O_2_). Cells were treated with 10 mM KBrO_3_ for 75 min and fixed in 4% PFA (Life Technologies/Thermo Fisher Scientific cat# 28908, Carlsbad, CA, USA).

#### 2.5.2. Immunofluorescence for R-Loops and FEN1

Fixed fibroblast cells were permeabilized (0.1% Triton-100 (Thermo Fisher Scientific, cat# 85111, Carlsbad, CA, USA), 0.05% Tween-20 (Sigma Aldrich, cat# P1379), 1 M glycine (Sigma Aldrich, cat# 50046, St. Louis, MO, USA) for 2 min at room temperature. To avoid the detection of non-specific signals resulting from single- and double-stranded RNA, cells were pretreated with single- and double-strand RNA nucleases RNase III (0.05 U/μL) (New England BioLabs, cat# M0245S, Ipswich, MA, USA) and RNase T1 (0.04 U/μL) (Sigma Aldrich, cat# R1003) in 1 x Shortcut buffer (New England BioLabs) with 20 mM MnCl_2_ (New England BioLabs) for 1 h at 37 °C. The negative control experiments were performed by digesting RNA:DNA hybrids in cells with 0.1 U/μL RNase H (NEB cat#M0297S) in RNaseH buffer (NEB) for 5 h at 37 °C. Cells were blocked in blocking buffer containing 3% BSA (Sigma Aldrich, cat# A7030), 3% donkey serum (Sigma Aldrich, cat# S30-M), 3% goat serum (Sigma Aldrich cat# S26-M) in PBS for 30 min at 37 °C and subjected to the incubation with the primary antibodies (1:500 dilution in 10% blocking buffer in PBS) for 1 h at 37 °C. Cells were washed twice with PBS (10 min each) and subjected to incubation with the secondary antibodies (1:1000 dilution in 10% blocking buffer in PBS) along with 0.1 μg/mL DAPI (Sigma Aldrich cat# D9542) for 30 min at 37 °C. Cells were washed twice with PBS (10 min each) and refixed (4% PFA for 20 min) before being covered in antifade (Immu-mount™, Fisher Scientific cat# 9990402) and imaged. Antibodies used for the experiments included S9.6 mouse anti-DNA:RNA hybrid antibody (MilliporeSigma, cat# MABE1095, Burlington, MA, USA), rabbit anti-FEN1 (AbCam, ab17994, Cambridge, MA, USA), donkey anti-mouse-Alexa488 (Jackson ImmunoResearch Laboratory Inc, cat# 715-545-150, West Grove, PA, USA), goat anti-Rb-Alexa546 (Thermo Fisher Scientific, cat# A11003). 

#### 2.5.3. Analysis of R-Loop Colocalization with FEN1 

Cells were imaged (on a Zeiss 710 confocal microscope, with a 100× 1.6 N/A oil lens, using an Airyscan detector). Images were analyzed using Fiji/ImageJ software with the JACoP pluggin [35]. Briefly, signals were thresholded with auto settings, and the nucleus of each cell was masked. The S9.6 signal was set as ImageA, and the FEN1 signal was set as ImageB. The Mander’s Overlap Coefficient [36,37] was then calculated for Fraction of A overlapping Fraction B. Ten cells per treatment of each cell type were analyzed.

## 3. Results

### 3.1. FEN1 Endonucleolytically Cleaves the RNA Strands in the RNA:DNA Hybrid Intermediates Formed during DNA Lagging Strand Processing

Since calf thymus FEN1 can cleave an RNA flap generated on a DNA lagging strand [20], we initially examined the endo- and exonucleolytic activity of human and yeast FEN1 on a nicked RNA, an RNA flap formed within an RNA strand, and an RNA flap attached to a DNA strand (Figure 1 and Figure 2). We found that human FEN1 at 1–10 nM made cleavage on the nicked RNA at multiple sites generating the products with 1 nt, 6 nt, 10 nt, and 17–19 nt (Figure 1A, lanes 3–5), whereas yeast FEN1 cleavage at 1–25 nM resulted in the products with 1 nt, 6 nt, and 7 nt (Figure 1D, lanes 2–5) indicating that FEN1 cleaved the RNA strand endonucleolytically. For the substrate containing a 10 nt-RNA flap within the RNA strand, both human and yeast FEN1 at 0.1–25 nM resulted in 10 nt cleavage products (Figure 1B,E, lanes 2–5), indicating that FEN1 made a cleavage at the junction between the RNA flap and annealed RNA region. FEN1 cleavage of a 19 nt-RNA flap attached to the downstream DNA strand at 0.1–25 nM resulted in a product with 21 nt (Figure 1C,F, lanes 2–5). The results indicated that FEN1 efficiently removed the RNA flap by making a cleavage within the DNA strand at the site 2 nt downstream from the RNA-DNA junction. We then examined if FEN1 5′-3′ exonuclease activity also cleaved RNA using the nick and flap RNA substrates radiolabeled at the 3′-end of the downstream strand (Figure 2). We found that human FEN1 at 1–10 nM on the nick RNA mainly generated the 21 nt-product along with a small amount of 30 nt-product (Figure 2A, lanes 3–5), indicating that FEN1 made the endonucleolytic cleavage at the RNA and RNA-DNA junction but failed to make 5′-3′ exonucleolytic cleavage on RNA. However, yeast FEN1 at 0.1–10 nM resulted in a product with 39 nt (Figure 2D, lanes 2–5), indicating that the enzyme only removed one nucleotide at the 5′-end of the RNA strand. For the 10 nt-RNA flap substrate, human FEN1 at 0.1–10 nM resulted in the cleavage products with 30 nt and 21 nt, whereas the same concentrations of yeast FEN1 mainly generated the 30 nt-cleavage products (Figure 2B,E, lanes 2–5). The results indicated that both human and yeast FEN1 efficiently cleaved the 10 nt-RNA flap within the RNA strand. However, human FEN1 also made the endonucleolytic cleavage at the junction between the RNA and DNA strands (Figure 2B, lanes 3–5). For the substrate containing a 19 nt-RNA flap attached to a DNA strand, human FEN1 cleavage only resulted in a 21 nt product (Figure 2C, lanes 2–5), indicating that the enzyme made a cleavage at the junction between the 19 nt-RNA flap and the annealed DNA strand. However, yeast FEN1 resulted in products with 21 nt, 20 nt, and 18 nt, indicating that the enzyme made the cleavage at 2 nt and 3 nt downstream of the RNA-DNA junction (Figure 2F, lanes 2–5). The results indicate that both human and yeast FEN1 employed its endonucleolytic activity to cleave the RNA strands. To validate that the RNA cleavage activity specifically resulted from FEN1 catalysis, we determined if the human FEN1 catalytic deficient mutant protein, FEN1D181A, can result in any cleavage of the RNA in the substrates (Appendix A). The results showed that no RNA cleavage products were generated by the FEN1 mutant protein (Appendix A), indicating that RNA cleavage activity specifically resulted from FEN1 catalysis. 

### 3.2. FEN1 Can Track down to the DNA Region to Remove an RNA Flap Endonucleolytically

We then asked if FEN1 also used its 5′-3′ exonuclease activity to continuously remove the RNA strand after the enzyme cleaved an RNA flap. We tested this possibility by examining human FEN1 cleavage activity on the substrate containing the 10 nt-RNA flap within RNA at various time intervals (Figure 3). The substrate was radiolabeled at the 3′-end of the downstream strand allowing the detection of the FEN1 exonucleolytic cleavage products. The results showed that FEN1 endonucleolytically cleaved the 10 nt-RNA flap leaving the 30 nt cleavage product at the time intervals of 1–60 min (Figure 3, lanes 3–8). At 5 min, a 21 nt-cleavage product was generated (Figure 3, lanes 7–8), indicating that FEN1 made the cleavage at the RNA-DNA junction after it removed the RNA flap. Moreover, we found that from 10 min, FEN1 started to exhibit its 5′-3′ exonuclease activity to cleave the downstream DNA strand (Figure 3, lanes 5–8). Quantification of the FEN1 endonucleolytic cleavage products showed that FEN1 endonucleolytic cleavage activity removed the 10 nt-RNA flap within 15 min along with a significant increase in the endonucleolytic cleavage at the RNA-DNA junction (Figure 3, the graph below the gel). The results indicated that FEN1 used its endonuclease rather than exonuclease activity to remove the RNA. The results further suggest that FEN1 removed the RNA flap leaving a nicked RNA for the enzyme to track down to the RNA-DNA junction and made the endonucleolytic cleavage, thereby removing the RNA strand.

### 3.3. FEN1 Is Recruited to R-Loops in Human Cells upon Oxidative DNA Damage and Cleaves the RNA in an R-Loop during BER

Since a recent study has implicated a role of FEN1 in processing R-loops in telomeres [11], and the non-template strand of an R-loop is susceptible to DNA base damage, we asked if FEN1 can employ endonucleolytic cleavage of RNA to remove the RNA strand in R-loops, facilitating the resolution of R-loops during BER. We initially examined if FEN1 can be recruited to R-loops upon oxidative DNA base damage induced by potassium bromate (KBrO_3_) in normal human fibroblasts and SETX helicase-deficient, AOA2 fibroblasts using immunofluorescence (Figure 4). We found that AOA2 fibroblasts had more R-loops accumulated than normal human fibroblasts (Appendix A) because of the deficiency of the SETX helicase [38]. We further demonstrated that in untreated normal fibroblasts and AOA2 fibroblasts, there was a basal level of colocalization of R-loops (green) and FEN1 (red) (~17%) detected in the nucleus of the fibroblasts (Figure 4A,B, the panels on the top). No significant difference in the colocalization between the normal and AOA2 fibroblasts was detected (Figure 4A). Upon the treatment of 10 mM KBrO_3_ that can specifically induce 8-oxoguanine (8-oxoG) in cells [39], the colocalization of R-loops and FEN1 in both types of cells was significantly increased by ~10% (Figure 4A,B, the panels in the middle). In cells treated with RNase H, only a non-specific signal was detected (Figure 4B, the panels at the bottom). The results indicated that FEN1 was recruited to R-loops upon the 8-oxoGs induced by KBrO_3,_ suggesting that FEN1 was recruited to process the RNA on R-loops during BER of oxidative DNA damage in normal and AOA2 fibroblasts. We then examined FEN1 cleavage activity on the RNA on an R-loop containing a 36 nt-RNA:DNA hybrid and an abasic lesion in the middle of the non-template strand (Figure 5). FEN1 cleavage of the RNA was determined with the substrate containing an intact abasic site (THF) or the substrate that mimics the intermediate generated by APE1 5′-incision of the abasic site in an R-loop, which contains a 3′- and 5′-DNA flap with an RNA:DNA hybrid (Figure 5). We found that FEN1 at 1–25 nM endonucleolytically cleaved the RNA strand from the R-loop substrate with an abasic site incised by 50 nM APE1 (Figure 5A, lanes 3–6). We confirmed that 50 nM APE1 converted almost all the R-loop substrate into the double-flap intermediate by incising the abasic site (THF) (Appendix A). Similar to its cleavage on the R-loop substrate, the same concentrations of FEN1 endonucleolytically cleaved the RNA strand on the double-flap R-loop intermediate (Figure 5B, lanes 3–6). For both substrates, the enzyme exhibited a unique cleavage pattern by initially cleaving 1 to 4 ribonucleotides endonucleolytically and subsequently removing a series of larger RNA fragments (Figure 5A,B, lanes 5–6). The results further suggest that FEN1 tracked along with the RNA strand to its 3′-end and made an endonucleolytic cleavage during BER of an R-loop. We then compared the FEN1 cleavage activity on the 5′-DNA flap with its cleavage on the RNA in the double-flap substrate (Figure 6). We found that the FEN1 cleavage of the 5′-flap is much faster than its cleavage on the RNA strand (Figure 5, compare the blue line with the black line). At 1 min, FEN1 at 5 nM cleaved 10% of the DNA flap but little RNA (Figure 6). At 5 min, FEN1 generated 30% of DNA flap cleavage products but only 5% of RNA cleavage products (Figure 6). At 15 min, FEN1 produced 50% of DNA flap cleavage products and 10% RNA cleavage products (Figure 6). The results indicate that FEN1 cleaved the 5′-DNA flap prior to its cleavage of the RNA strand in the R-loop during BER. 

### 3.4. APE1 3′-5′ Exonuclease Cleaves RNA in an R-Loop during BER

As one of the key BER enzymes, APE1 can incise the 5′-end of an abasic site using its endonucleolytic cleavage activity, creating a 1 nt-gap as the substrate for pol β [40]. APE1 also possesses its 3′-5′ exonuclease activity [40] that can remove a mismatched nucleotide at the 3′-end of the upstream strand of a one-nucleotide gap [41]. Furthermore, it is shown that APE1 can also make endo- and exonucleolytic cleavage on RNA to exhibit its ribonuclease activity [42,43,44]. We reason that APE1 exoribonuclease may also cleave the RNA strand in an R-loop during BER. To test this, we examined the APE1 cleavage activity on the RNA strand of the double-flap substrate with the 36 nt-RNA and the substrate containing the RNA strand with a 3′-flap alone (Figure 7). We found that APE1 at 50 nM only removed one ribonucleotide from the 3′-end of the RNA strand on the double-flap substrate with low efficiency (Figure 7A, lane 5). APE1 lower than 50 nM failed to cleave the RNA strand (Figure 7A, lanes 2–4). However, APE1 efficiently removed ribonucleotides from the 3′-end of the RNA on the substrate with the 3′-flap alone (Figure 7B). In contrast, APE1 catalytic deficient mutant APE1H309A protein failed to cleave the RNA on the substrates (Appendix A). The results indicated that APE1 progressively cleaved the ribonucleotides from the 3′-end of the RNA in an R-loop during BER, and its ribonuclease activity was significantly stimulated after the 5′-DNA flaps were removed by FEN1. To further confirm this, we compared the relative efficiency of APE1 exoribonuclease activity on the RNA in the substrate containing the double-flap and the 3′-flap at various time intervals (Figure 8). We found that APE1 exhibited inefficient 3′-5′ exoribonuclease activity on the RNA strand on the double-flap substrate with only 10% cleavage products generated in 60 min (Figure 8, red line). However, APE1 exoribonuclease efficiently cleaved the RNA in the substrate containing the 3′-flap alone (Figure 8, black line), with 80% of products generated in 60 min. The results indicate that APE1 3′-5′ exoribonuclease cleaved the RNA on an R-loop after FEN1 removed the 5′-DNA flap suggesting the coordination between FEN1 DNA flap cleavage and APE1 exoribonuclease in processing the RNAs on an R-loop during BER. 

### 3.5. FEN1 and APE1 Cleavage of RNA Coordinates to Facilitate BER in an R-Loop

To further determine if FEN1 and APE1 cleavage on RNA can coordinate to promote the resolution of an R-loop through BER, we reconstituted BER on the R-loop substrates in the presence of increasing concentrations of FEN1 or APE1 (Figure 9). We found that increasing concentrations of FEN1 (1–25 nM) led to the increasing amount of repaired products with the presence of 5 nM pol β, 10 nM LIG I in the presence and absence of 50 nM APE1 (Figure 9A,B, lanes 4–7). Increasing concentrations of APE1 at 5–50 nM in the presence of a high concentration of FEN1 at 25 nM did not alter the amount of the repaired product (Figure 9C, lanes 4–7). However, since APE1 is much more abundant than FEN1 in human cells, the results suggest that the large amount of APE1 can employ its 3′-5′ exonuclease activity to cleave RNA in R-loops through coordinating with a limited amount of FEN1 in cells. The results indicate that FEN1 and APE1 cleavage of the RNA in an R-loop coordinated to resolve the R-loop via BER. The necessity of BER on the R-loop was further supported by that the results showing that FEN1 failed to cleavage RNA in the R-loop without DNA base damage (Appendix A).

## 4. Discussion

In this study, we identified a unique role of FEN1 RNA processing in resolving an R-loop through BER. We found that human and yeast FEN1 endonucleolytically cleaved a nicked RNA and an RNA flap (Figure 1 and Figure 2). We further demonstrated that FEN1 removed an RNA flap and then tracked along with the RNA strand to the RNA-DNA junction to cleave the RNA (Figure 3). We then showed that FEN1 was recruited to R-loops in normal human fibroblasts and SETX-deficient fibroblasts, and its recruitment was facilitated by oxidative DNA damage, 8-oxoG (Figure 4). Further analysis of FEN1 cleavage on the RNA strand during BER on an R-loop showed that FEN1 initially cleaved the downstream 5′-DNA flap and then endonucleolytically cleaved the RNA strand (Figure 5 and Figure 6). Moreover, we found that APE1 exhibited an efficient 3′-exoribonucleolytic cleavage of the RNA strand in an R-loop by coordinating with the FEN1 cleavage of the 5′-DNA flap (Figure 7 and Figure 8). We showed that the RNA cleavage activity of FEN1 and APE1 facilitated the removal and resolution of an R-loop via BER (Figure 9). Our results supported a model during which oxidative stress induces DNA base lesions such as 8-oxoGs and abasic sites on the non-template strand of an R-loop. 8-oxoG DNA glycosylase 1 (OGG1) removes 8-oxoGs and generates abasic sites on the non-template strand. APE1 incises the AP sites converting the R-loop into a double-flap intermediate with an RNA:DNA hybrid. Subsequently, FEN1 removes the downstream 5′-DNA flap leaving an intermediate with a 3′-DNA flap and an RNA:DNA hybrid. FEN1 then uses its endonucleolytic cleavage activity to track along the RNA strand and remove the RNA from its 5′-end, whereas APE1 3′-exoribonucleolytically cleaves the RNA from the 3′-end of the RNA strand. The coordinated cleavage of RNA by FEN1 and APE1 exoribonuclease results in the resolution of the R-loop and a DNA gap filled in by pol β gap-filling synthesis. Finally, LIG I seals the nick and complete the BER leading to the resolution of the R-loop (Figure 10).

Early work from the Bambara group shows that calf thymus FEN1 can cleave an RNA flap [20], indicating that mammalian FEN1 also possesses ribonuclease activity. The work by Stevens has further demonstrated that human FEN1 can endonucleolytically cleave mRNA and rRNA at the junction of RNA flaps, hairpins, and bubbles [21]. A recent study also points out a potential role of FEN1 in processing RNA:DNA hybrid and preventing telomere fragility during DNA leading strand synthesis [11]. However, the mechanisms by which FEN1 ribonuclease processes RNA and its biological significance remain unknown. Here, we provide the first mechanistic insight into human FEN1 ribonuclease activity by showing that the enzyme cleaves RNA endonucleolytically (Figure 1 and Figure 2). We found that FEN1 removed an RNA flap and then tracked along the RNA strand to the RNA-DNA junction or DNA region to make an endonucleolytic cleavage (Figure 1, Figure 2 and Figure 3) suggesting the unique mechanism for FEN1 to cleave RNA. We further demonstrated that FEN1 employed its RNA cleavage activity to process the RNA strand in an R-loop via BER (Figure 4, Figure 5 and Figure 6) indicating the important cellular function for FEN1 to maintain genome stability and integrity in cells. Our results further demonstrated that FEN1 5′-endoribonuclease coordinated with APE1 3′-exoribonuclease to remove the RNA strand in an R-loop via BER of a base lesion on the non-template strand leading to the resolution of the R-loop (Figure 7, Figure 8 and Figure 9). Thus, our results revealed a unique mechanism for FEN1 to remove RNAs leading to resolution of R-loops through BER. The roles of FEN1 in RNA metabolism and miRNA biogenesis and their impact on genome maintenance need to be elucidated in the future.

The Hickson group identified the cleavage of RNAs by APE1 in the 1990s [42]. Later studies have shown that APE1 endoribonuclease can incise the 5′-end of an abasic site in mRNAs and RNA in a site-specific manner leading to RNA degradation [45,46]. The enzyme also exhibits endo- and exoribonuclease activity on an ssRNA [44]. In this study, for the first time, we found that APE1 3′-exoribonuclease but not endoribonuclease activity progressively cleaved the RNA strand of an RNA:DNA hybrid in an R-loop during BER (Figure 7 and Figure 8). We showed that the 3′-exoribonuclease activity of APE1 was dependent on the removal of the downstream 5′-DNA flap by FEN1 (Figure 7 and Figure 8), demonstrating the functional coordination between the BER enzymes in removing the RNA strand during BER in an R-loop. It is possible that the 5′-DNA flap in the double-flap intermediate formed during BER in an R-loop inhibited the binding of APE1 to the 3′-end of the RNA strand, thereby preventing the enzyme from making a ribonucleotide cleavage. 

Our study further indicates that BER may serve as a unique channel to resolve R-loops through FEN1 endonucleolytic cleavage of DNAs and RNAs in coordination with APE1 3′-exoribonuclease (Figure 10). Several studies have implicated the role of FEN1 in R-loop processing [10,30]. However, it is unknown when and how the enzyme can perform its flap cleavage on R-loops. Here, we showed that FEN1 colocalized with R-loops in cells upon oxidative DNA base damage (Figure 4), suggesting that FEN1 was recruited to R-loops to process the structures in a DNA damage-dependent manner. This is supported by the fact that FEN1 is absent in the RNA:DNA hybrid interactome in HeLa cells in the absence of DNA damage [47] and by our results that FEN1 failed to cleavage the RNA strand on the R-loop without a DNA base damage (Appendix A). Interestingly, a recent study from the Cheung group has shown that several key BER enzymes, pol β, FEN1, and some BER cofactors are absent in the list of the proteins that bind to RNA:DNA hybrid of R-loops formed in B-cells [48]. Since scheduled R-loops are essential for class switching recombination in B-cells for generating antibodies, the finding suggests that the BER pathway is excluded from the resolution of scheduled R-loop but specifically employed in resolving nonscheduled R-loops upon DNA base damage. 

Our results showed that DNA base lesions that occurred on the non-template strand of an R-loop were efficiently removed through BER. This, in turn, resulted in the ssDNA breaks on the non-template strand, converting the R-loop into a double-flap with an RNA:DNA hybrid that was removed by FEN1 DNA flap cleavage and endoribonuclease activity along with APE1 3′-exoribonuclease activity. Since it is implicated that a DNA glycosylase, SMUG1 [49] and BER cofactor, proliferating cell nuclear antigen (PCNA), and poly(ADP-ribose) polymerase 1 (PARP1) are also involved in RNA metabolism [50] and R-loop processing [47], it is conceivable that the BER cofactors may also coordinate with FEN1 and APE1 leading to the efficient removal of the RNA on R-loops and their resolution via BER. The molecular mechanisms underlying the resolution of R-loops through BER needs to be elucidated in the future. 

## 5. Conclusions

Our study has demonstrated that FEN1 can process RNA endonucleolytically. FEN1 endonucleolytic flap cleavage coordinates with APE1 exoribonuclease activity to remove RNA strands in RNA:DNA hybrid, thereby resolving an R-loop through BER.

## Figures and Tables

**Figure 1 genes-14-00098-f001:**
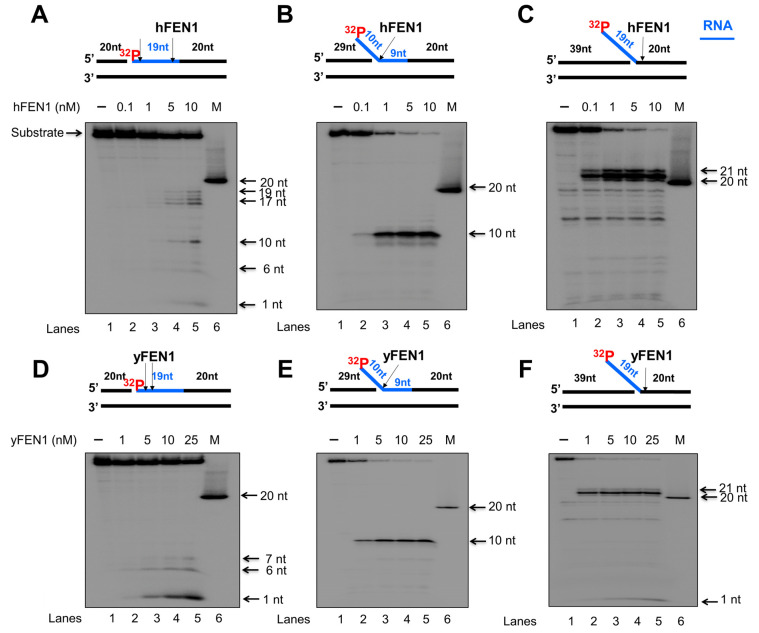
FEN1 endonucleolytic cleavage of an RNA strand in RNA:DNA hybrids. Human and yeast FEN1 cleavage activity on a RNA strand of RNA:DNA hybrid was measured by incubating different concentrations of FEN1 (0.1–25 nM) with the substrates (25 nM) that mimics the intermediates generated during DNA lagging strand processing. Substrates were radioactively labeled at the 5′-end of the downstream strand containing an RNA (blue) and DNA (black) fragment on the same strand. (**A**) Human FEN1 cleavage on the nicked RNA:DNA hybrid substrate. (**B**) Human FEN1 cleavage on the RNA:DNA hybrid substrate intermediate containing a 9 nt-RNA flap. (**C**) Human FEN1 cleavage on the substrate containing a 19 nt-RNA flap attached to a duplex DNA. (**D**) Yeast FEN1 cleavage on the substrate containing a nicked RNA:DNA hybrid. (**E**) Yeast FEN1 cleavage on the substrate containing a 9 nt-RNA flap in a RNA/DNA hybrid. (**F**) Yeast FEN1 cleavage on the substrate containing a 19 nt-RNA flap attached to a duplex DNA. Lane 1 represents substrate only. Lanes 2–6 represent the reaction mixture with increasing concentrations of FEN1. “M” represents a size marker of 20 nt. Substrates and products were separated in a 15% urea-denaturing polyacrylamide gel and detected by phosphorimagery. All experiments were performed in triplicate.

**Figure 2 genes-14-00098-f002:**
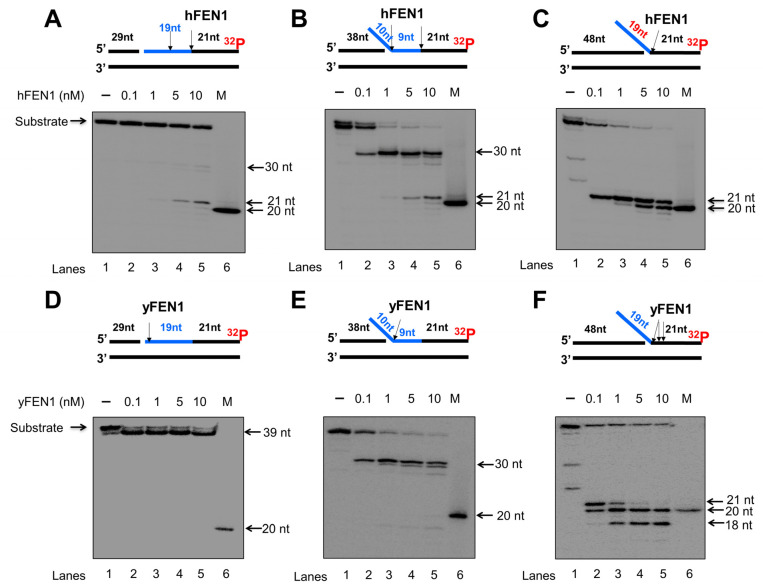
FEN1 exonucleolytic cleavage of RNA:DNA hybrids. Human and yeast FEN1 exonucleolytic cleavage activity on the RNA strand in the context of RNA/DNA hybrids was measured by incubating various concentrations of FEN1 (0.1–10 nM) with the substrates (25 nM) at 37 °C for 30 min. Substrates were ^32^P-labeled at the 3′-end of the downstream DNA strand containing a RNA (blue). (**A**) Human FEN1 cleavage on the RNA on the substrate containing a nicked RNA:DNA hybrid. (**B**) Human FEN1 cleavage on the substrate containing a 9 nt-RNA flap of generated from a RNA:DNA hybrid. (**C**) Human FEN1 cleavage on the substrate containing a 19 nt RNA flap attached to a duplex DNA. (**D**) Yeast FEN1 cleavage on the substrate containing a nicked RNA:DNA hybrid. (**E**) Yeast FEN1 cleavage on the substrate containing a 9 nt-RNA flap generated within RNA:DNA hybrid. (**F**) Yeast FEN1 RNA cleavage on duplex DNA containing a 19 nt-RNA flap. Lane 1 represents substrate only. Lanes 2–6 represent the reaction mixture with increasing concentrations of FEN1. “M” represents a size marker of 20 nt. Substrates and products were separated in a 15% urea-denaturing polyacrylamide gel and detected by phosphorimager. The experiments were performed in triplicate.

**Figure 3 genes-14-00098-f003:**
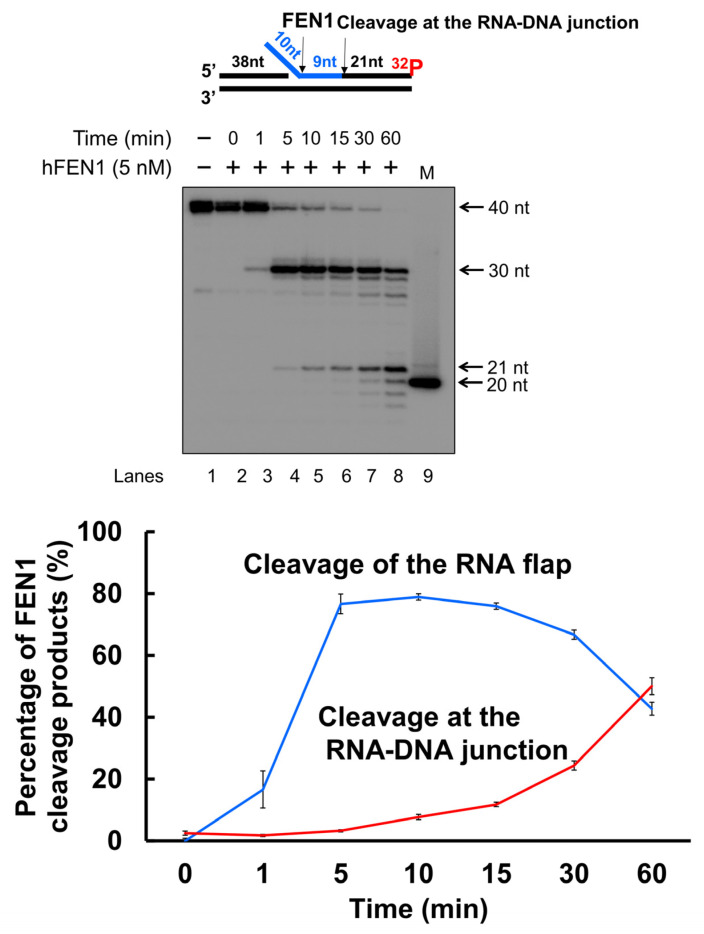
FEN1 cleavage of RNA and DNA in an RNA/DNA hybrid at different time intervals. To compare the efficiency of FEN1 cleavage on the RNA flap and RNA-DNA junction, FEN1 (5 nM) cleavage on the RNA (blue) and DNA strand of the substrate (25 nM) containing a 10 nt RNA flap was measured at different time intervals (0–60 min). The results were shown in the gel (top panel). Lane 1 represents substrate only. Lane 2–8 represents the reaction mixture with FEN1 cleavage products generated from 0–60 min. Lane 9 is a 20 nt size marker. Quantification of the FEN1 cleavage products is shown in the bottom panel. The line in blue indicates FEN1 cleavage on the RNA flap. The line in red indicates FEN1 cleavage at the RNA-DNA junction. The substrate was ^32^P-labeled at the 3′-end of the downstream strand of the substrate. Substrate and products were separated in a 15% urea-denaturing polyacrylamide gel and detected by phosphorimagery. The experiments were performed in triplicates.

**Figure 4 genes-14-00098-f004:**
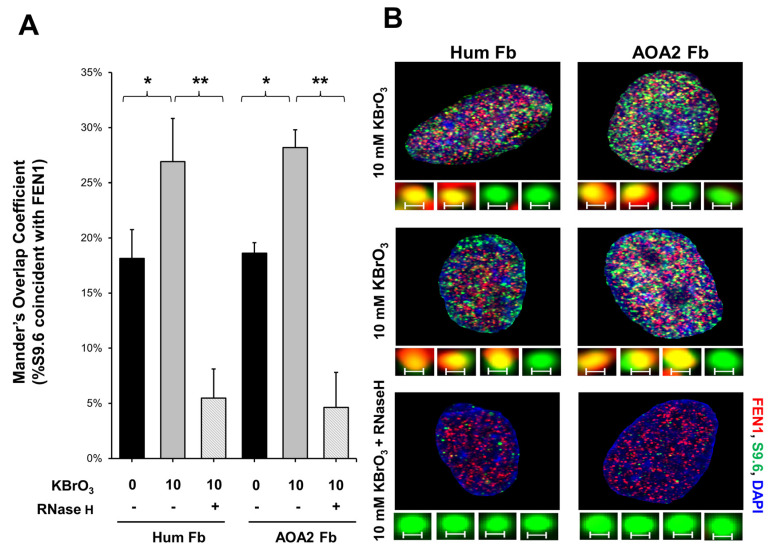
FEN1 recruitment to R-loops in human fibroblasts following oxidative DNA base damage. (**A**) The Mander’s Overlap Coefficient (percentage of the anti-R-loop antibody, S9.6 signal that overlaps with FEN1) indicated that the colocalization of R-loops and FEN1 (FEN1 recruitment to R-loops) increased after oxidative DNA damage (10 mM KBrO_3_ treatment for 75 min) in both normal human fibroblasts and in senataxin deficient AOA2 fibroblasts. Error bars are standard error of the mean. “*” and “**” indicate statistical significance (*p* < 0.05 and *p* < 0.005, respectively from Student *t*-test, 2 tailed, homoscedastic). (**B**) Representative images of nuclei of both cell types, either treated or non-treated in the presence of absence of RNase H and of (insets) non-colocalized (green) and colocalized (yellow) foci of FEN1 (red) and S9.6 (RNA-DNA) (green). Scale bar = 0.2 μm.

**Figure 5 genes-14-00098-f005:**
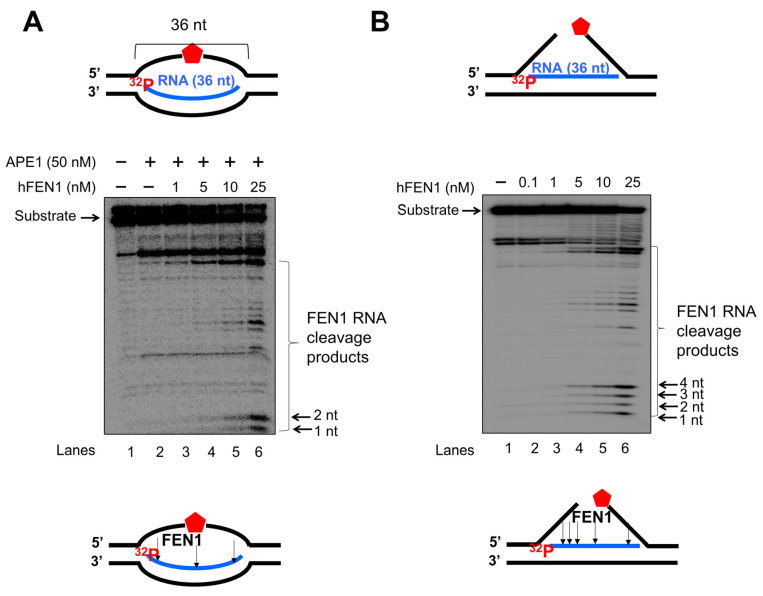
FEN1 cleavage of RNA on R-loop intermediates during BER. FEN1 cleavage on the RNA strand on an R-loop during BER was examined using the substrate containing an abasic site in the non-template strand (**A**) or substrate containing a 3′- and 5′-flap (**B**) that mimics the intermediate generated from the 5′-incision of an abasic site on the non-template strand of an R-loop. (**A**) FEN1 cleavage (1–25 nM) on the RNA strand of the R-loops in the presence of 50 nM APE1. Lane 1 represents substrate only. Lane 2 represents APE1 cleavage of the abasic site on the non-template strand of the R-loop. Lanes 3–6 represents FEN1 cleavage activity on the RNA strand at different concentrations. (**B**) FEN1 cleavage (0.1–25 nM) on the RNA strand of the double-flap intermediate generated by BER in R-loop. Lane 1 represents substrate only. Lanes 2–6 represents the reaction mixture with FEN1 at different concentrations. Substrates 25 nM were ^32^P-labeled at the 5′-end of the RNA strand. Substrates and products were separated in a 15% urea-denaturing polyacrylamide gel and detected by phosphorimagery. The experiments were done in triplicate.

**Figure 6 genes-14-00098-f006:**
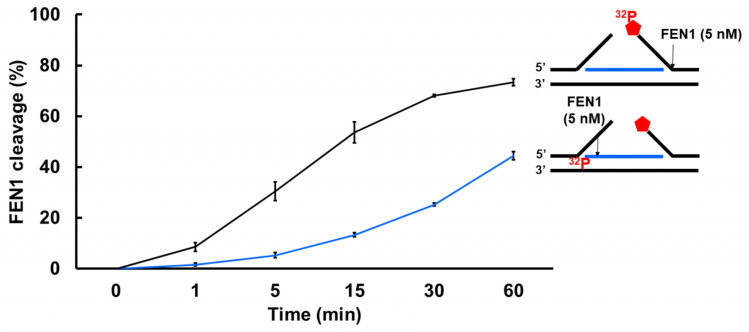
FEN1 cleavage on the DNA flap and RNA in an R-loop intermediate at different time intervals. The efficiency of FEN1 cleavage on the DNA flap and RNA in an R-loop intermediate was examined by determining the activity of FEN1 (5 nM) on the double-flap substrate (25 nM) containing a RNA:DNA hybrid at different time intervals (0–60 min). FEN1 cleavage on the 5′-DNA flap was measured using the substrate that was ^32^P-labeled at the 5’-end of the downstream DNA flap. The percentage of FEN1 DNA flap cleavage products was plotted against different time intervals (line in black). FEN1 cleavage activity on the RNA strand in the R-loop substrate was measured using the substrate labeled with ^32^P at the 5′-end of the RNA. The percentage of FEN1 RNA cleavage products in different time intervals was plotted (line in blue). All experiments were performed in triplicate.

**Figure 7 genes-14-00098-f007:**
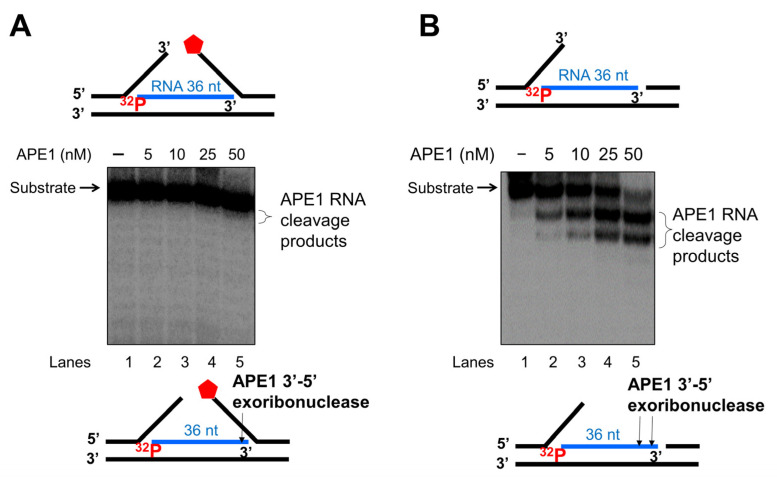
APE1 3′-exonucleolytic cleavage of RNA on R-loops intermediates. APE1 3′-exonucleolytic cleavage on RNA on R-loops was examined by incubating different concentrations of APE1 (5–100 nM) with the substrate (25 nM) that mimics the R-loop intermediate containing double-flaps or a 3′-flap alone at 37 °C for 15 min. Substrates were ^32^P-labeled at the 5′-end of the RNA strand of the substrate. (**A**) The activity of APE1 cleavage of the RNA strand in the double-flap R-loop substrate. (**B**) APE1 exonucleolytic cleavage of the RNA strand on the R-loop substrate containing a 3′-flap alone. Lane 1 represents the substrate only. Lane 2–6 represents the reaction mixture with increasing concentrations of APE1. Substrates and products were separated in a 15% urea-denaturing polyacrylamide gel and detected by phosphorimagery. The experiments were performed in triplicate.

**Figure 8 genes-14-00098-f008:**
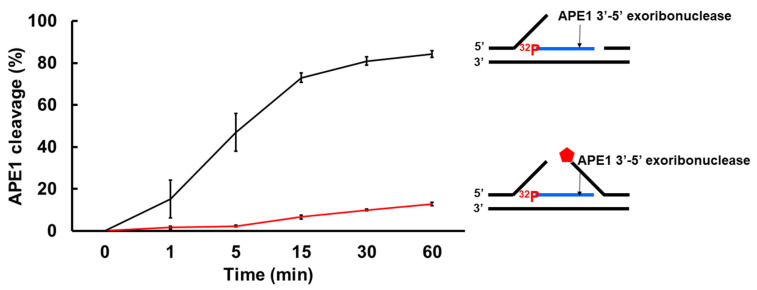
APE1 exonucleolytic cleavage of RNA is stimulated by the absence of the downstream 5′-DNA flap in an R-loop. The efficiency of APE1 exonucleolytic cleavage on the RNA strand on the R-loop intermediate with and without a 5′-flap was examined using the substrates labeled with ^32^P at the 5′-end of the RNA strand of the substrates. Substrates (25 nM) were incubated with 25 nM APE1 at 37 °C at different time intervals. The percentage of APE1 3′-exonucleolytic cleavage products were plotted against the time. APE1 exonucleolytic cleavage on the RNA strand in the R-loop substrate with 3′-flap alone was in black, whereas its cleavage activity on the RNA in the R-loop double-flap substrate was in red. The experiments were performed in triplicate.

**Figure 9 genes-14-00098-f009:**
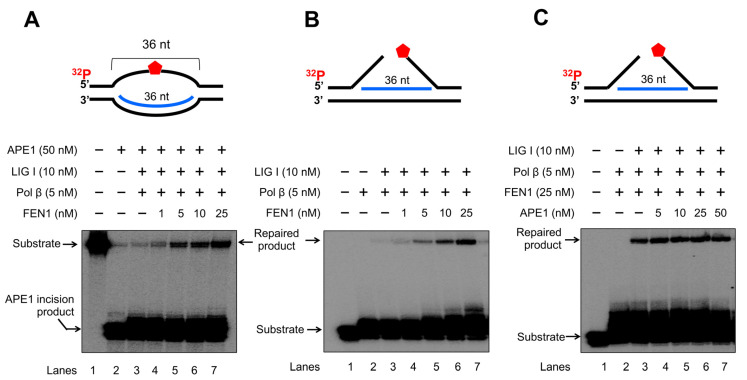
FEN1 and APE1 coordinate to promote the resolution of R-loops during BER. The coordination between FEN1 endonucleolytic cleavage activity and APE1 exonucleolytic activity in resolving R-loops through BER was determined by reconstituting BER with 25 nM R-loop substrates with an abasic site on the non-template strand and R-loop double-flap substrate. Enzymes and substrates were incubated at 37 °C for 30 min. (**A**) BER was reconstituted with 25 nM R-loop substrate containing an abasic site on the non-template strand and different concentrations of FEN1 (1–25 nM) in the presence of 50 nM APE1, 5 nM pol β and 10 nM LIG I. Lane 1 represents substrate only. Lane 2 represents the results from APE1 cleavage on the R-loopsubstrate. Lane 3 illustrates the results of FEN1 flap cleavage on the R-loop substrate. Lanes 4–7 represent the results of reconstituted BER in the presence of increasing concentrations of FEN1. (**B**) BER was reconstituted with 25 nM R-loop double-flap substrate in the presence of 5 nM pol β and 10 nM LIG I. Lane 1 represents substrate only. Lane 2 indicates the reaction mixture with pol β and the substrate. Lane 3 illustrates the reaction mixture with pol β and LIGI and the substrate. Lanes 4–7 represent the reaction mixture with increasing concentrations of FEN1. (**C**) BER was reconstituted with the R-loop double-flap substrate with increasing concentrations of APE1 in the presence of 5 nM pol β, 25 nM FEN1, and 10 nM LIG I. Lane 1 represents substrate only. Lane 2 illustrates the reaction mixture with pol β and FEN1. Lane 3 indicates the reaction mixture with FEN1, pol β, and LIG I. Lanes 4–7 represent the reaction mixture with increasing concentrations of APE1 in the presence of pol β, FEN1 and LIG I. Substrates and products were separated in a 15% urea-denaturing polyacrylamide gel and detected by phosphorimagery. The experiments were performed in triplicate.

**Figure 10 genes-14-00098-f010:**
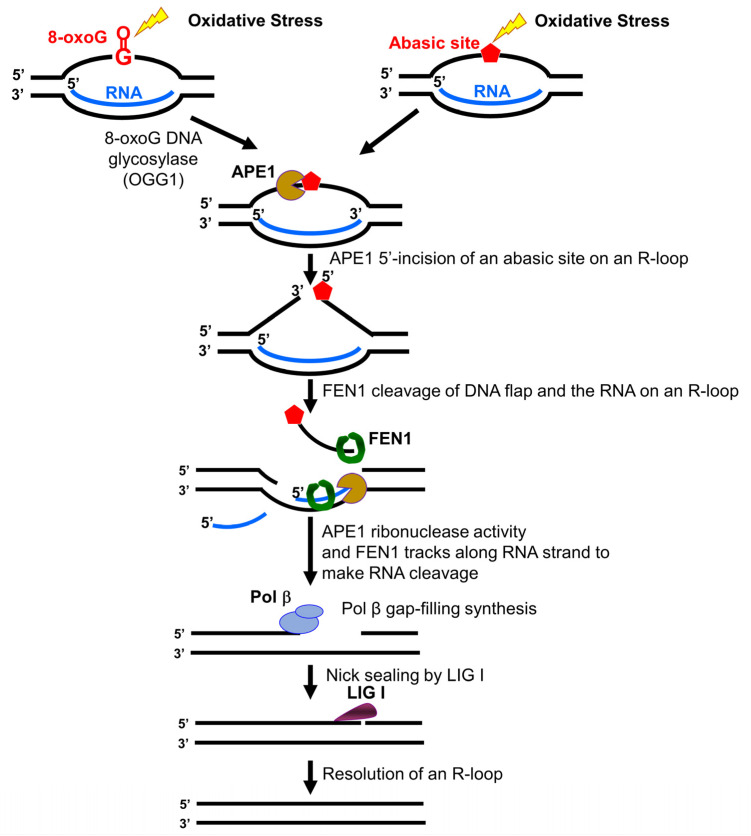
FEN1 and APE1 coordinate to cleave RNA and resolve an R-loop via BER. Oxidative stress induces DNA base lesions such as 8-oxoGs and abasic sites in the non-template strand of an R-loop. 8-oxoG DNA glycosylase (OGG1) removes the 8-oxoG leaving an abasic site that is subsequently 5′-incised by APE1. This incises the non-template strand converting the R-loop into a double-flap with a RNA:DNA hybrid. The downstream 5′-DNA flap is removed by FEN1. This allows FEN1 and APE1 to access the RNA strand in the R-loop. FEN1 endonucleolytically cleaves RNA from the 5′-side, whereas APE1 exonucleolytically removes ribonucleotides from the 3′-end of the RNA. The removal of the RNA by the coordinated cleavage activity of FEN1 and APE1 allows the reannealing of the upstream flap to the template strand leaving a gap. Pol β then fill in the gap generating a nick that is sealed by LIG I. This leads to the completion of BER of the base lesions and the resolution of the R-loop.

## Data Availability

All the data and images will be deposited into Zenodo linked with ORCID to enhance the public accessibility of the results. A Zenodo personal access token for Yuan Liu was created for the deposit.

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
