# Peer review of "Flap Endonuclease 1 Endonucleolytically Processes RNA to Resolve R-Loops through DNA Base Excision Repair"

_genes, 2022, doi:10.3390/genes14010098_

Round 1

Reviewer 1 Report

The authors showed that the enzyme FEN1 can cleave RNA endonucleolytically and provided an mechanistic insight into human FEN1 ribonuclease activity . Specifically, they found that FEN1 removed an RNA flap and then tracked along the RNA strand to the RNA-DNA junction or DNA region to make an endonucleolytic cleavage. This is the unique mechanism for FEN1 to cleave RNA. They also demonstrated that FEN1 employed its RNA cleavage activity to process the RNA strand in an R-loop via BER. This observation indicates the and important cellular function for FEN1 to maintain genome stability and integrity in cells. These results further suggested that FEN1 5'-endoribonuclease coordinated with APE1 3'-exoribonuclease to remove the RNA strand in an R-loop via BER of a base lesion on the non-template strand leading to the resolution of the R-loop (Figures 7-9). Together, this work reveals a unique mechanism for FEN1 to remove RNAs leading to resolution of R-loops through BER. The experimental designs are elegant, and the writting is good. The work is excellent. However, one figure (Figure 1) missed in the manuscript.   Major concerns: 1) Does the depletion of Fen1 cause accumulation in R-loop in the cells?   Minor concerns: 1) Figure 1 is missed.

Author Response

Dear Cathryn:

Thank you very much for your decision letter on October 26 concerning our manuscript. We appreciate the constructive comments from the reviewers that help improve our manuscript significantly. We are very grateful for the opportunity to resubmit the revised manuscript entitled “Flap endonuclease 1 endonucleolytically processes RNA to resolve R-loops through DNA base excision repair” for publication in the Genes. Below please find our responses to address the reviewer’s concerns point-by-point.

Reviewer 1

1) Major concerns: Does the depletion of Fen1 cause accumulation in R-loop in the cells?  

Response: We thank the reviewer’s insightful comment. A recent study has shown that the depletion of FEN1 in human cells results in the deficiency of R-loop processing, thereby promoting the accumulation of R-loops induced by topoisomerase I cleavage complex (TOP1ccs) and reducing DNA strand breaks resulting from R-loop processing (Cristini et al.  2019, Cell Reports 28, 3167–3181). The results further suggest a crucial role of FEN1 in resolving R-loops. We discussed this in the “Introduction” of the revised manuscript (page 2, lines 85-87, red).

2) Minor concerns: 1) Figure 1 is missed.

Response: We thank the reviewer’s careful reading. We apologize for the system errors that occurred while submitting and uploading our manuscript. We corrected the errors by providing the revised manuscript that includes Figure 1 and all other figures and supplementary Table S1 and figures.

Reviewer 2 Report

The manuscript, entitled “Flap endonuclease 1 endonucleolytically processes RNA to resolve R-loops through DNA base excision repair”, shows the novel function of FEN1 to resolve R-loop. As the mechanism of R-loop resolution is remarked currently and the important evidence are being reported, this manuscript might be interesting for the related-area researcher. Although authors show several data about possible roles for R-loop resolution in vitro, the data are not enough to prove the existence of function for R-loop resolution in vivo (in cells) about following concerns. Furthermore, I have another serious problem to evaluate the manuscript, because the manuscript for the reviewer pdf lacks Figure 6 and supplementary Figures 1~5. Such lacks should be improved in revision. Therefore, I think that the manuscript needs major revision.

Major points

1) Although the authors used potassium bromate in Figure 4, dose potassium bromate induce R-loop? In order to show the appropriateness of this agents, the authors have to show the induction of R-lop (e. g. S9.6 antibody foci, not merged foci with FEN1). If potassium bromate cannot induce (increase) R-loop, the author had better use another proper reagent for Figure 4.

2) Although the authors insist the recruitment of FEN1 to R-loop from Figure 4, the author just observed the overlapping. Usually, PLA technology with S9.6 antibody is used, the recruitment of protein factor to recruitment (accumulation). Please try for Figure 4.

3) I understand that senataxin (SETX) is one of important factor for R-loop resolving. Therefore, it is expected that AOA3 cells increase R-loop. Please investigate it as related experiments with above 1) and 2).

4) If FEN1 functions in R-loop under oxidative stress condition, FEN1-defective cells might be sensitive to oxidative stress-inducing reagents. Please try viability assay (colony formation assay) with potassium bromate or proper reagent in FEN1-kockdowned cells. And, the results of doble-knockdown cells (FEN1 and APE1) might be interesting.

5) From Supplementary Figure 5 (I don’t have and don’t’ see it), the authors insist that FEN1 failed to cleavage the RNA strand on the R-loop without a DNA base damage. I think that the authors also had better prove the unnecessary for R-loop without DNA base damage in vivo (in cells). Aphidicolin or camptothecin are known to induced R-loop independent of oxidative damage, please try whether the FEN1 doesn’t recruit to R-loop induced with these reagents with immunofluorescent experiments.

Minor point

6) What kinds is the used human fibroblast (in Figure)? Please show exact cell name in Material and Methods section. If possible, please add the reference.

7) I cannot find “3. Results”. And, it is difficult to find each section title in Results. Please improve them in the revision.

Author Response

Dear Cathryn:

Thank you very much for your decision letter on October 26 concerning our manuscript. We appreciate the constructive comments from the reviewers that help improve our manuscript significantly. We are very grateful for the opportunity to resubmit the revised manuscript entitled “Flap endonuclease 1 endonucleolytically processes RNA to resolve R-loops through DNA base excision repair” for publication in the Genes. Below please find our responses to address the reviewer’s concerns point-by-point.

Reviewer 2

“…the manuscript for the reviewer pdf lacks Figure 6 and supplementary Figures 1~5. Such lacks should be improved in revision.”

Response: We thank the reviewer for pointing out the issues. We apologize for the missing of Figure 6 and Supplementary Table S1, and Supplementary Figures S1-S5 due to the system error that occurred during the submission and uploading of our manuscript. We provided Figure 6 and all supplementary data in the revised manuscript.

Major points

1) Although the authors used potassium bromate in Figure 4, dose potassium bromate induce R-loop? In order to show the appropriateness of this agents, the authors have to show the induction of R-lop (e. g. S9.6 antibody foci, not merged foci with FEN1). If potassium bromate cannot induce (increase) R-loop, the author had better use another proper reagent for Figure 4.

Response: We thank the reviewer’s interesting point. However, R-loops are generated by gene transcription (Garcia-Muse, T. and Aguilera, A. (2019) R Loops: From Physiological to Pathological Roles. Cell, 179, 604-618) but not induced by DNA damaging agents such as potassium bromate.

Thus, the suggested potassium bromate experiments do not address the question of FEN1 processing of R-loops in our study.

2) Although the authors insist the recruitment of FEN1 to R-loop from Figure 4, the author just observed the overlapping. Usually, PLA technology with S9.6 antibody is used, the recruitment of protein factor to recruitment (accumulation). Please try for Figure 4.

Response: We thank the reviewer’s suggestion of using the Proximity Ligation Assay (PLA) as an alternative approach to detect the recruitment of FEN1 to R-loops. However, a recent study has shown that PLA can even detect the nonspecific interactions between human proteins with the nonhuman green fluorescence protein and IgGs (Alsemarz, A, Lasko, P, Fagotto, F (2018). Limited significance of the in situ proximity ligation assay. BioRxiv preprint first posted online Sep. 9, 2018; doi: http://dx.doi.org/10.1101/411355). The findings indicate that PLA can lead to false positive results and misinterpretation of the experimental results. Thus, the technical limitations of PLA prevented us from using the method in this study.

 3) I understand that senataxin (SETX) is one of important factor for R-loop resolving. Therefore, it is expected that AOA3 cells increase R-loop. Please investigate it as related experiments with above 1) and 2).

Response: As the reviewer pointed out, our results did show that AOA2 fibroblasts exhibited a significantly higher level of R-loops than normal fibroblasts (Supplementary Figure S2)(page 7,  lines 299-302, red). Also, as we already addressed the points raised by the reviewer, the suggested experiments do not address how FEN1 processes R-loops or are not technically feasible for addressing FEN1 recruitment to R-loops.

4) If FEN1 functions in R-loop under oxidative stress condition, FEN1-defective cells might be sensitive to oxidative stress-inducing reagents. Please try viability assay (colony formation assay) with potassium bromate or proper reagent in FEN1-kockdowned cells. And, the results of doble-knockdown cells (FEN1 and APE1) might be interesting.

Response: We appreciate the reviewer’s interesting points. Although the suggested experiments are important for testing the roles of FEN1 and APE1 in mediating oxidative DNA damage repair, which have already been well-defined in the DNA repair field, the experiments cannot address the specific roles of FEN1 in processing R-loops in our study.

5) From Supplementary Figure 5 (I don’t have and don’t’ see it), the authors insist that FEN1 failed to cleavage the RNA strand on the R-loop without a DNA base damage. I think that the authors also had better prove the unnecessary for R-loop without DNA base damage in vivo (in cells). Aphidicolin or camptothecin are known to induced R-loop independent of oxidative damage, please try whether the FEN1 doesn’t recruit to R-loop induced with these reagents with immunofluorescent experiments.

Response: We apologize for missing the supplementary results, including Supplementary Figure S5, due to the system errors that occurred during the uploading of our manuscript and supplementary Figures and Table S1. We corrected the errors in the revised manuscript.

Our results showed that without base damage and DNA strand breaks, FEN1 failed to cleave the RNA strand in the R-loop (Supplementary Figure 5).

We thank the reviewer’s suggestion of using aphidicolin or camptothecin to induce R-loops for FEN1 recruitment. However, it should be noted that aphidicolin is a DNA polymerase inhibitor that leads to replication fork stalling, which in turn can cause DNA damage, such as double-strand DNA breaks. Camptothecin is a DNA topoisomerase inhibitor that induces single-strand break, which can lead to FEN1 accumulation on the strand breaks. Thus, the use of the reagents can cause DNA damage and cannot address whether FEN1 can cleave RNA on R-loops in the absence of DNA damage on R-loops in vivo.

Minor point

6) What kinds is the used human fibroblast (in Figure)? Please show exact cell name in Material and Methods section. If possible, please add the reference.

Response: We appreciate the reviewer’s careful reading. According to the reviewer’s suggestion, we provided the information about the primary fibroblasts provided by Dr. Kenneth Fischbeck from the National Institute of Neurological Disorders and Stroke/National Institutes of Health (page 3, lines 103-105, red). The cells were generous gifts from Dr. Fischbeck. The information on fibroblasts has not been published.

7) I cannot find “3. Results”. And, it is difficult to find each section title in Results. Please improve them in the revision.

Response: We thank the reviewer’s careful reading and helpful suggestions. We include “3. Results” and the numbers for the subheadings in the “Results” (page 4, lines 179-180, page 6, line 257, line 290, page 8, line 349, page 9, line 390, red).

Reviewer 3 Report

The manuscript by Eduardo E. Laverde et al., “Flap endonuclease 1 endonucleolytically processes RNA to resolve R-loops through DNA base excision repair” describes characterization of the FEN1-dependent degradation of RNA in RNA:DNA heteroduplex oligonucleotides as a part of BER process in R-loops. The data presented in the manuscript indicated that FEN1 can endonucleolytically remove an RNA flap up to its junction with DNA in RNA:DNA heteroduplexes and that FEN1 can degrade RNA to resolve an R-loop after incision of the damaged non-template strand of this R-loop during BER. Although this work contains some intriguing results, the presented data are insufficient to support some of the conclusions.

General comments:

1.       Figure 1 is missing.

2.       The authors wrote “We further demonstrated that FEN1 was recruited to R-loops in human normal fibroblasts and senataxin-deficient (AOA2) fibroblasts, and its R-loop recruitment was significantly increased by oxidative DNA damage”. This conclusion is based on the colocalization analysis of FEN1 and the R-loop in Figure 4, however, it is well known that anti-DNA-RNA Hybrid [S9.6] antibody performs much better for immunoprecipitation than immunofluorescence assay and can create some non-specific signals in cells. Authors should run a control experiment with additional RNase H treatment to confirm the specific R-loop detection.

3.   The authors state that FEN1 specifically uses its endonucleolytic cleavage activity to remove RNA in the R-loop (Figure 5). However, in a similar experiment from their previous work (ref. 22, Fig. 7), only the exonuclease activity of FEN1 on RNA in the R-loop can be observed. What is the reason for this discrepancy? Authors should label the 3' end of the RNA oligonucleotide to test for the 5' exonuclease activity of FEN1.

4.    The authors wrote “Increasing concentrations of APE1 (5 nM-50 nM) in the presence of a high concentration of FEN1 at 25 nM moderately altered the amount of the repaired product (Figure 9C, lanes 4-7)”. I don't see any difference in lanes 4-7, so a role for APE1 3-exonuclease activity in R-loop RNA cleavage is too speculative.

Specific comments:

 “3. Results” title is missing

Line 99. Point missing after “(30)”

Line 175. senataxin, not senataxn

Legend of Figure 3. Please revise “To compare the efficiency of FEN1 cleavage on the RNA and DNA on the substrates….”

Line 499. “the important”, not “the and important”

Author Response

Dear Cathryn:

Thank you very much for your decision letter on October 26 concerning our manuscript. We appreciate the constructive comments from the reviewers that help improve our manuscript significantly. We are very grateful for the opportunity to resubmit the revised manuscript entitled “Flap endonuclease 1 endonucleolytically processes RNA to resolve R-loops through DNA base excision repair” for publication in the Genes. Below please find our responses to address the reviewer’s concerns point-by-point.

Reviewer 3

 General comments:

  1. Figure 1 is missing.

Response: We apologize for missing Figure 1 in the submitted manuscript due to the system errors that occurred during the submission and uploading of the files. We have provided Figure 1, Figure 6, Supplementary Table S1, and Supplementary Figures S1-S5 in the revised manuscript.

  1. The authors wrote “We further demonstrated that FEN1 was recruited to R-loops in human normal fibroblasts and senataxin-deficient (AOA2) fibroblasts, and its R-loop recruitment was significantly increased by oxidative DNA damage”. This conclusion is based on the colocalization analysis of FEN1 and the R-loop in Figure 4, however, it is well known that anti-DNA-RNA Hybrid [S9.6] antibody performs much better for immunoprecipitation than immunofluorescence assay and can create some non-specific signals in cells. Authors should run a control experiment with additional RNase H treatment to confirm the specific R-loop detection.

Response: We thank the reviewer’s helpful suggestion. It is reported that non-specific signals can be detected using the S9.6 antibody in normal immunofluorescence staining experiments (ref. 1, 2). To avoid the detection of the non-specific signals resulting from single- and double-strand RNA, in our experiments, we pretreated the cells with single- and double-strand RNA nucleases (RNase III and RNase T1) as described in Material and Methods and shown as follows.

To avoid the detection of non-specific signals resulting from single- and double-stranded RNA, cells were pretreated with single- and double-strand RNA nucleases RNase III (0.05 U/μL) (New England BioLabs, cat# M0245S, Ipswich, MA) and RNase T1 (0.04 U/μL)(Sigma Aldrich, cat# R1003) in 1 x Shortcut buffer (New England BioLabs) with 20 mM MnCl2 (New England BioLabs) for 1 hr at 37oC (page 4, lines 160-163, red)

Reference

1. Smolka JA, Sanz LA, Hartono SR, Chedin F. Recognition of cellular RNAs by the S9.6 antibody creates pervasive artefacts when imaging RNA:DNA hybrids. BioRxiv. 2020. doi: 10.1101/2020.01.11.902981
2. Nadel J, Athanasiadou R, Lemetre C, Wijetunga NA, P OB, Sato H, Zhang Z, Jeddeloh J, Montagna C, Golden A, Seoighe C, Greally JM. RNA:DNA hybrids in the human genome have distinctive nucleotide characteristics, chromatin composition, and transcriptional relationships. Epigenetics Chromatin. 2015;8:46. doi: 10.1186/s13072-015-0040-6. PubMed PMID: 26579211; PMCID: PMC4647656.

  1.  The authors state that FEN1 specifically uses its endonucleolytic cleavage activity to remove RNA in the R-loop (Figure 5). However, in a similar experiment from their previous work (ref. 22, Fig. 7), only the exonuclease activity of FEN1 on RNA in the R-loop can be observed. What is the reason for this discrepancy? Authors should label the 3' end of the RNA oligonucleotide to test for the 5' exonuclease activity of FEN1.

Response: We appreciate the reviewer’s careful reading. In Figure 7 of our previous study (ref 22), the human FEN1 cleavage products contained at least one GAA or CAG repeat, indicating that FEN1 also exhibited endonucleolytic cleavage on the RNA strand in the trinucleotide repeat R-loops. To examine if FEN1 5’-3’ exonuclease activity also cleaved RNA, the RNA-containing substrates were radiolabeled at the 3’-end of the downstream strand (Figure 2). The results showed that human FEN1 at 1 nM-10 nM on the nick RNA mainly generated the 21 nt-product along with a small amount of 30 nt-product (Figure 2A, lanes 3-5) indicating that FEN1 made the endonucleolytic cleavage at the RNA and RNA-DNA junction but failed to make 5’-3’ exonucleolytic cleavage on RNA. We clarified this in the revised manuscript (page 5, lines 236-240, red).

  1.   The authors wrote “Increasing concentrations of APE1 (5 nM-50 nM) in the presence of a high concentration of FEN1 at 25 nM moderately altered the amount of the repaired product (Figure 9C, lanes 4-7)”. I don't see any difference in lanes 4-7, so a role for APE1 3-exonuclease activity in R-loop RNA cleavage is too speculative.

Response: We appreciate the reviewer’s constructive comment. We explained the implication of the results for a significant role of APE1 3’-5’ exonuclease activity in cells by cleaving RNA through coordinating with FEN1, as shown below.

Increasing concentrations of APE1 (5 nM-50 nM) in the presence of a high concentration of FEN1 at 25 nM moderately altered the amount of the repaired product (Figure 9C, lanes 4-7). Since APE1 is much more abundant than FEN1 in cells, the results suggest that the large amount of APE1 can employ its 3’-5’ exonuclease activity to cleave RNA in R-loops by coordinating with a limited amount of FEN1 in cells (page 9, lines 404-414, red)

Specific comments:

  1. “3. Results” title is missing

Response: We thank the reviewer for careful reading. The “Results” title is included along with the number of subheadings (page 4, lines 179-180, page 6, line 257, line 290, page 8, line 349, page 9, line 390, red).

  1. Line 99. Point missing after “(30)”

Response: We thank the reviewer’s careful reading. We add the “.” after (30) (page 2, line 85, red).

  1. Line 175. senataxin, not senataxn

Response: As the reviewer pointed out, we corrected the misspelling error (page 3, line 149, red).

  1. Legend of Figure 3. Please revise “To compare the efficiency of FEN1 cleavage on the RNA and DNA on the substrates….”

Response: As the reviewer suggested, we revised the sentence in the legend of Figure 3, shown as the following.

“To compare the efficiency of FEN1 cleavage on the RNA flap and RNA-DNA junction, FEN1 (5 nM) cleavage on the RNA (blue) and DNA strand of the substrate (25 nM) containing a 10 nt RNA flap was measured at different time intervals (0 min-60 min).”(page 6, the legend of Figure 3, red)

  1. Line 499. “the important”, not “the and important”

 Response: As the reviewer pointed out, we corrected the error (page 11, lines 467, red).

Round 2

Reviewer 2 Report

Although I don't agree with authors' responses about 1) 2) 3), I understand the authors' views to our concerns. I don't request more revision.

Author Response

Reviewer 2

Although I don't agree with authors' responses about 1) 2) 3), I understand the authors' views to our concerns. I don't request more revision.

Response: We thank the reviewer’s help and support.

Reviewer 3 Report

The manuscript has improved slightly, but unfortunately the authors have not responded correctly to my second and fourth comments:

2. “Response: We thank the reviewer’s helpful suggestion. It is reported that non-specific signals can be detected using the S9.6 antibody in normal immunofluorescence staining experiments (ref. 1, 2). To avoid the detection of the non-specific signals resulting from single- and double-strand RNA, in our experiments, we pretreated the cells with single- and double-strand RNA nucleases (RNase III and RNase T1) as described in Material and Methods and shown as follows.”

As RNA make up the majority of the S9.6 IF signal in cells, the treatment used was obligatory but not sufficient. As stated in the cited article by Smolka et al. (now published in Journal of Cell Biology doi: 10.1083/jcb.202004079 ): “…These pretreatments (RNase III and RNase T1), coupled with RNase H controls, should help investigators use S9.6 in a thoroughly controlled manner that can account for the capacity of S9.6 to recognize RNA.” Without negative RNase H controls, one cannot claim that those treatments have eliminated all non-specific signals!

4. The authors once again insist that “Increasing concentrations of APE1 (5 nM-50 nM) in the presence of a high concentration of FEN1 at 25 nM moderately altered the amount of the repaired product (Figure 9C, lanes 4-7)”. This is incorrect, Figure 9C shows no effect of APE1 at any concentration used.

Author Response

Reviewer 3

1) As RNA make up the majority of the S9.6 IF signal in cells, the treatment used was obligatory but not sufficient. As stated in the cited article by Smolka et al. (now published in Journal of Cell Biology doi: 10.1083/jcb.202004079 ): “…These pretreatments (RNase III and RNase T1), coupled with RNase H controls, should help investigators use S9.6 in a thoroughly controlled manner that can account for the capacity of S9.6 to recognize RNA.” Without negative RNase H controls, one cannot claim that those treatments have eliminated all non-specific signals!

Response: We thank the reviewer’s constructive comment. As the reviewer suggested, we conducted the control experiments by treating cells using RNase H. The experiments were described on page 4, lines 163-164 in red. The results showed that in cells treated by RNase H only a low background level of R-loops was detected indicating that the S9.6 signals represented R-loops (Figure 4B, the panels at the bottom)(page 7, lines 299-301, red).

2) The authors once again insist that “Increasing concentrations of APE1 (5 nM-50 nM) in the presence of a high concentration of FEN1 at 25 nM moderately altered the amount of the repaired product (Figure 9C, lanes 4-7)”. This is incorrect, Figure 9C shows no effect of APE1 at any concentration used.

Response: We thank the reviewer for pointing this out. We corrected this by modifying the sentences to “Increasing concentrations of APE1 at 5 nM-50 nM in the presence of a high concentration of FEN1 at 25 nM did not alter the amount of the repaired product (Figure 9C, lanes 4-7). However, since APE1 is much more abundant than FEN1 in human cells, the results suggest that the large amount of APE1 can employ its 3’-5’ exonuclease activity to cleave RNA in R-loops through coordinating with a limited amount of FEN1 in cells.” (page 9, lines 385-393, red)

Round 3

Reviewer 3 Report

The manuscript is greatly improved and the changes address my prior comments. I only have a minor comment to the authors :

The S9.6 signal after RNase H treatment does not represent “background level of R-loops” but a nonspecific signal. Please correct the corresponding sentence: The results showed that in cells treated by RNase H, only a low background level of S9.6 signal was detected indicating that the majority of S9.6 signals represented R-loops (Figure 4B, the panels at the bottom).

Author Response

Reviewer 3

The S9.6 signal after RNase H treatment does not represent “background level of R-loops” but a nonspecific signal. Please correct the corresponding sentence: The results showed that in cells treated by RNase H, only a low background level of S9.6 signal was detected indicating that the majority of S9.6 signals represented R-loops (Figure 4B, the panels at the bottom)

Response: We thank the reviewer's helpful suggestion. As the reviewer suggested, we changed the sentence to "In cells treated with RNase H, only a non-specific signal was detected (Figure 4B, the panels at the bottom)." as indicated using tracking (page 7, lines 299-301, red).